# A2A Receptor Dysregulation in Dystonia DYT1 Knock-Out Mice

**DOI:** 10.3390/ijms22052691

**Published:** 2021-03-07

**Authors:** Vincenza D’Angelo, Mauro Giorgi, Emanuela Paldino, Silvia Cardarelli, Francesca R. Fusco, Ilaria Saverioni, Roberto Sorge, Giuseppina Martella, Stefano Biagioni, Nicola B. Mercuri, Antonio Pisani, Giuseppe Sancesario

**Affiliations:** 1Department of Systems Medicine, Tor Vergata University of Rome, 00133 Rome, Italy; dangelo@med.uniroma2.it (V.D.); sorge@uniroma2.it (R.S.); martella@med.uniroma2.it (G.M.); mercurin@med.uniroma2.it (N.B.M.); 2Department of Biology and Biotechnology “Charles Darwin”, Sapienza University of Rome, 00185 Rome, Italy; mauro.giorgi@uniroma1.it (M.G.); silvia.cardarelli@uniroma1.it (S.C.); ilaria.saverioni@gmail.com (I.S.); stefano.biagioni@uniroma1.it (S.B.); 3IRCCS Santa Lucia Foundation, 00179 Rome, Italy; e.paldino@hsantalucia.it (E.P.); f.fusco@hsantalucia.it (F.R.F.); 4IRCCS Mondino Foundation, 27100 Pavia, Italy; pisani@uniroma2.it; 5Department of Brain and Behavioral Sciences, University of Pavia, 27100 Pavia, Italy

**Keywords:** A2A, cAMP, A2A mRNA, dystonia, DYT1, basal ganglia, D2

## Abstract

We aimed to investigate A2A receptors in the basal ganglia of a DYT1 mouse model of dystonia. A2A was studied in control Tor1a+/+ and Tor1a+/− knock-out mice. A2A expression was assessed by anti-A2A antibody immunofluorescence and Western blotting. The co-localization of A2A was studied in striatal cholinergic interneurons identified by anti-choline-acetyltransferase (ChAT) antibody. A2A mRNA and cyclic adenosine monophosphate (cAMP) contents were also assessed. In Tor1a+/+, Western blotting detected an A2A 45 kDa band, which was stronger in the striatum and the globus pallidus than in the entopeduncular nucleus. Moreover, in Tor1a+/+, immunofluorescence showed A2A roundish aggregates, 0.3–0.4 μm in diameter, denser in the neuropil of the striatum and the globus pallidus than in the entopeduncular nucleus. In Tor1a+/−, A2A Western blotting expression and immunofluorescence aggregates appeared either increased in the striatum and the globus pallidus, or reduced in the entopeduncular nucleus. Moreover, in Tor1a+/−, A2A aggregates appeared increased in number on ChAT positive interneurons compared to Tor1a+/+. Finally, in Tor1a+/−, an increased content of cAMP signal was detected in the striatum, while significant levels of A2A mRNA were neo-expressed in the globus pallidus. In Tor1a+/−, opposite changes of A2A receptors’ expression in the striatal-pallidal complex and the entopeduncular nucleus suggest that the pathophysiology of dystonia is critically dependent on a composite functional imbalance of the indirect over the direct pathway in basal ganglia.

## 1. Introduction

The clinical features of dystonia are relatively well-defined, produced by the ill-timed activation of agonist-antagonist muscles, and presenting either as focal or generalized. Conversely, the pathological mechanisms underlying such a heterogeneous group of movement disorders remain elusive [1,2,3]. Unlike symptomatic dystonia secondary to histological damages affecting basal ganglia, no neuropathologic correlates are detectable at microscopic levels for primary dystonia [4,5].

A common form of primary early onset generalized dystonia is caused by 3 bp deletion (GAG) in the coding region of the TOR1A (DYT1) gene, which results in a defective protein called torsinA, whose role in dystonia pathology is unclear [6]. In animal models for DYT1 dystonia, multiple lines of evidence revealed the impairment of dopamine receptor type 2 (D2 receptor), with D2 downregulation, sparse D2 synapses, reduced coupling between the D2 receptor and its cognate G proteins, as well as the loss of D2 dependent electrophysiological inhibition and severely altered synaptic plasticity in medium spiny neurons and cholinergic interneurons in the striatum [7,8,9,10,11,12,13,14,15,16,17,18]. Therefore, the striatum and D2 receptors have been considered central in the cellular pathomechanism underlying DYT1 dystonia [15].

Noteworthy, the deficit in D2 receptor-mediated transmission can be counteracted by A2A receptor pharmacological antagonism in the striatum of a genetic mouse model of DYT1, and also of DYT11 and DYT25 dystonia, suggesting that an imbalance in D2/A2A functions may represent a convergent pathogenetic mechanism [11,19,20]. Moreover, a peculiarity of D2 receptors is their selective co-localization with A2A receptors in striatal cholinergic interneurons [21,22], and mainly in the sub-population of striatal medium spiny GABAergic neurons containing enkephalin, which are known to lead the indirect striatal-pallidal pathway [23,24]. D2 and A2A receptors in enkephalin neurons form complex heteromers and exert reciprocal antagonistic interaction, inhibiting or stimulating the second messenger cyclic adenosine monophosphate (cAMP) synthesis, respectively [25,26].

Therefore, besides the dysfunction of D2 receptors, A2A receptors may be of interest in dystonia pathophysiology for several reasons. In this paper, we aimed to investigate whether the D2 receptor dysfunction is coupled with any change in the A2A receptor expression in the basal ganglia of a DYT1 mouse model. We report that opposite to the downregulation of D2 receptors in the striatum, an up-regulation of the A2A receptors occurs in the striatal-pallidal complex, whereas A2A down-regulation occurs in the entopeduncular nucleus of a DYT1 dystonia mouse model.

## 2. Results

### 2.1. Expression of A2A Receptors

We first quantified the presence of A2A receptors by Western blot on proteins extracted from the striatum, globus pallidus and entopeduncular nucleus of Tor1a+/+ and Tor1a+/−. A single specific band was detected in all areas of the basal ganglia approximately at 45 kDa, which corresponds to the migration level of A2A receptor monomers (Appendix A). In Tor1a+/+, a higher A2A expression was detected in the striatum that was not significantly different from the globus pallidus, whereas the lowest A2A expression was detected in the entopeduncular nucleus (*p* < 0.001). Moreover, Western blotting analysis revealed a significant increase in A2A receptor levels in the striatum (*n* = 11, df = 1, F = 5.189, *p* = 0.034) and globus pallidus (*n* = 9, df = 1, F = 6.791, *p* = 0.020), but a significant decrease in A2A expression in the entopeduncular nucleus (*n* = 8, df = 1, F = 7.629, *p* = 0.016) of mutant Tor1a+/− compared to control Tor1a+/+ mice (Figure 1).

### 2.2. Cyclic AMP Levels

We then investigated whether the changes in A2A expression in the basal ganglia of Tor1a+/− mice consistently affected second messenger cAMP content in tissue extracts. Actually, the cAMP level was significantly increased in the striatum (df = 1, F = 4.828, *p* < 0.05), but was unchanged in the globus pallidus (df = 1, F = 0.151, *p* > 0.5) and in the entopeduncular nucleus (df = 1, F = 0.403, *p* > 0.5) of Tor1a+/− mice (*n* = 7) compared with Tor1a+/+ (*n* = 7) (Figure 2).

### 2.3. Expression of A2A mRNA

Real-time PCR amplification of A2A mRNA was performed in the two groups of animals to investigate A2A changes at the transcription level in basal ganglia. The primers, which amplified a specific mRNA fragment (140 bp), have shown that the A2A mRNA expression level in the caudate–putamen of Tor1a+/+ (*n* = 6) is much higher than in the globus pallidus (*n =* 8) (Figure 3A), whereas in the entopeduncular nucleus its expression is under the detection level (data not shown). The pattern and relative intensity of A2A mRNA expression detected in the caudate–putamen of Tor1a+/− mice (*n* = 8) were similar compared to the caudate–putamen of Tor1a+/+ mice (Figure 3B). Surprisingly, a significant increase in A2A mRNA expression was detected in the globus pallidus of Tor1a+/−mice (*n =* 8) compared to Tor1a+/+ mice (Figure 3C).

### 2.4. A2A Immunofluorescence in Confocal Microscopy

To allow a precise morphological definition of the A2A receptor aggregates on the striatal-pallidal complex in control and mutant mice, we performed a fluorescence staining, followed by detection with confocal microscopy. A better understanding of A2A receptors’ subcellular distribution came out in images acquired using 63× oil immersion objective (1.4 numerical aperture) with an additional digital zoom factor (1×–1.5×–2×). The immuno-fluorescent signal appeared extremely specific without background staining, showing A2A positive small grains with elliptical or roundish shape about 0.3–0.4 μm in diameter, isolated or contiguous, composing irregular clusters of size variable in the different areas of the basal ganglia of Tor1a+/+ mice (Figure 4A,C,E).

Noteworthy, in Tor1a+/+ mice, A2A small grains appeared to be diffusely covering the neuronal compartments of the striatum, globus pallidus and entopeduncular nucleus, uniformly distributed in the neuropil, whereas grains were rare and almost absent on the cell nuclei and in striatal axonal bundles (Figure 4A,C,E). Moreover, among the basal ganglia regions, the number of A2A receptors per microscopic field was much higher in the striatum and globus pallidus than in the entopeduncular nucleus.

In Tor1a+/− mice, A2A receptor grains appeared either increased in number in the striatum and globus pallidus, or reduced in number in the entopeduncular nucleus (Figure 4B,D,F), compared with control Tor1a+/+ (Figure 4A,C,E). Moreover, the distribution pattern of A2A positive fluorescent grains was clearly different in the basal ganglia of Tor1a+/− mice, wherein A2A receptor positive grains appeared as tiny spots uniformly distributed in the neuropil, but also clustered on neuronal bodies of the striatum and globus pallidus (Figure 4B,D), unlike in Tor1a+/+ mice (Figure 4A,C).

Densitometric analysis confirmed a higher density of the number of A2A receptors per microscopic field in the striatum (df = 1, F = 14.039, *p* = 0.013) and in the globus pallidus (df = 1, F = 22.859, *p* = 0.003), and their lower density in the entopeduncular nucleus (df = 1, F = 9.031, *p* = 0.024) of Tor1a+/− (*n* = 4), compared with correspondent basal ganglia areas of the Tor1a+/+ mice (*n* = 4) (Figure 5).

While the numbers of A2A receptors were either increased in the striatal-pallidal complex or decreased in the entopeduncular nucleus, their size appeared smaller in all the basal ganglia structures of TOR1a+/−. A representative number of A2A positive grains were randomly selected from the basal ganglia of control and mutant mice: the mean perimeters per A2A positive grain in TOR1a+/+ versus TOR1a+/− were, respectively, (two tailed T test): in the striatum, 2.06 ± 0.19 > 1.62 ± 0.08 SEM μm (*n* = 13, t = 2.077 df = 24, *p* = 0.04); in the globus pallidus, 2.342 ± 0.14 > 1.378 ± 0.06 SEM μm (*n* = 15, t = 6.088 df = 28, *p* = 0.0001); and in the entopeduncular nucleus, 4.938 ± 0.39 > 1.61 ± 0.08 SEM μm (*n* = 14, t = 8.238, df = 26, *p* = 0.0001). However, the fluorescence intensity of the A2A positive grains was comparable in the correspondent areas of TOR1a+/+ versus TOR1a+/− (data not shown).

### 2.5. Colocalization of A2A in Cholinergic Neurons

Finally, we evaluated whether any change also occurs in the expression of A2A receptors in the striatal choline-acetyltransferase (ChAT) positive cholinergic interneurons of mutant Tor1a+/− mice. Double-labeling fluorescence was obtained, with ChAT reactive large neuronal bodies and thick primary dendrites marked in red (Figure 6B,E), and A2A receptor positive grains marked in green in the neuropil (Figure 6A,D). In the merged ChAT/A2A images, A2A receptors appeared as yellow grains uniformly diffused on ChAT neuronal bodies and thick dendrites (Figure 6C,F), but absent in the white matter bundles appearing as black holes in the microscopic field (Figure 6A,C,D,F), confirming the specificity of A2A receptor staining on neurons. Compared to Tor1a+/+ mice (Figure 6C), in Tor1a+/− the A2A receptor grains (Figure 6F) appeared markedly more numerous on the surface of ChAT neurons (yellow grains) and in the surrounding neuropil (green grains).

The overlapping areas of A2A receptors on striatal ChAT neuronal bodies were 36.37 + 2.92 SEM μm^2^ (neuronal bodies *n* = 12 from 3 mice) in Tor1a+/+ mice, and 47.44 + 2.98 SEM μm^2^ (neuronal bodies *n* = 13 from 3 mice) in Tor1a+/− mice (unpaired two-tailed T test, t = 2.645, df = 23, *p* = 0.014).

## 3. Discussion

In our study, different techniques demonstrate specific changes in A2A receptors in the basal ganglia circuits of Tor1a+/− mice, with increased or decreased expression, respectively, in the striatal-pallidal complex and entopeduncular nucleus. We briefly discuss the limits of our morpho-chemical study, and the possible relevance of A2A changes to the pathophysiology of dystonia in this DYT1 animal model.

The high specificity of the polyclonal antibody was demonstrated in the Western blot analysis, with a selective band corresponding to the 45 kDa migration level of A2A receptor monomers [27], and by the well-defined immunofluorescent reaction product, configuring the sharp morphology of A2A receptor aggregates without surrounding diffuse background staining. Moreover, our Western blot analysis and immunohistochemical method were extremely sensitive, also detecting the low level of A2A receptors in the entopeduncular nucleus of control Tor1a+/+ mice, and even their decrease in this nucleus, contemporary to the opposite changes of A2A expression in the contiguous striatal-pallidal complex on the same tissue section of Tor1a+/− mice. 

### 3.1. Morphological Characteristics of A2A Receptor Aggregates 

In previous studies with electron microscopy, the A2A labeling in the striatum was most commonly localized in dendrites and dendritic spines, in fewer amounts in axon terminals and in very low levels in neuronal soma and glia, whereas in the external globus pallidus (GP), the A2A labeling was mainly presynaptic [28,29]. In our study, the immunofluorescence A2A aggregates appeared arranged in a homogenous spot, with a diameter between 0.3 and 0.4 μm, which approaches the dimensions and morphology of synaptic complexes [30]. However, we cannot define the subcellular localization of the A2A aggregates at synaptic and extra synaptic levels since we cannot detect anything but the fluorescent A2A signals, losing the surrounding subcellular structures configuring the synaptic complex as detected in electron microscopy. Instead, the advantage of the immuno-fluorescence technique and confocal microscopy with high power objective is that they can allow at a time detection of A2A aggregates, approaching synaptic size at cellular level, as well as their distribution in the tissue in a relatively large microscopic field.

### 3.2. Distribution of A2A Aggregates

The high distribution of adenosine A2A receptors in the striatum and globus pallidus of control Tor1a+/+ mice and their selective increase in the same areas of Tor1a+/− mice suggest a major role of A2A within the indirect pathway for basal ganglia physiology and dystonia pathophysiology. However, it is worth noting that besides the selective postsynaptic localization of A2A receptors on striatal medium spiny enkephalin neurons in the indirect pathway [25], a morphological segregation of A2A receptors has been demonstrated at presynaptic levels in glutamatergic terminals that make synapses on striatal medium spiny neurons of the direct pathway, where they exert a selective facilitator modulation of cortico-striatal neurotransmission [31]. Further studies should investigate the localization of A2A receptors at synaptic and extra-synaptic levels in dystonic mice, clarifying the distinct changes of A2A receptors in basal ganglia sub-regions.

Our immuno-fluorescence technique and confocal microscopy with high power objective demonstrate the increased distribution density of A2A in Tor1a+/− mice in the neuropil, likely targeting dendrites and spines in the striatum and globus pallidus, as well as on the soma of the striatal projecting neurons, striatal cholinergic interneurons, and neurons in the globus pallidus. The widespread increase in A2A receptors in the striatal-pallidal complex may have functional relevance, affecting acetylcholine release in the striatum and regulating neuronal excitability and plasticity on cortico-striatal glutamatergic terminals, on GABAergic projecting neurons, and on GABAergic terminals in the globus pallidus [21,25,31,32].

Moreover, the light A2A expression generally observed in the entopeduncular nucleus of Tor1a+/+ mice [33] and its significant decrease in Tor1a+/− mice, detected with different techniques, cannot be considered functionally irrelevant, and this suggests that A2A receptors in the direct pathway could also be involved in basal ganglia function and in dystonia pathophysiology. We hypothesize that the increased expression of A2A in the striatum and in the globus pallidus is likely involved in increased inhibitory GABA input at striatal-pallidal synapses demonstrated in globus pallidus in DYT1 dystonia [32], disinhibiting the subthalamic nucleus and its excitatory glutamatergic input to the entopeduncular nucleus [34]. In this scenario, the reduced A2A expression in the entopeduncular nucleus may be viewed, at least in part, as an adaptive mechanism to regulate increased glutamatergic input from the subthalamic nucleus to basal ganglia output. Future studies should investigate the elaborated role of pre- and postsynaptic A2A receptors in the glutamatergic synapses of the direct- and indirect pathway neurons, respectively, on GABAergic medium spiny neurons’ soma and dendrites in the striatum, and of their terminals in the entopeduncular nucleus and globus pallidus in the pathophysiology of dystonia. While the numbers of A2A positive dots were either increased or decreased, respectively, in the striatal-pallidal complex and entopeduncular nucleus of Tor1a+/− mice, at the same time, the size of A2A positive dots was significantly decreased in all three structures. Such a decrease in the size of A2A receptor aggregates could be associated with the contemporary decrease in the D2 synapses [18], and the consequent deficiency in A2A/D2 colocalization and the formation of heteromers, at least in the striatum. The functional relevance of such A2A microstructural change should be checked in the direct and indirect pathway since, in general, the efficiency of neuronal connectivity is directly related to the synapses’ size [35]. Further studies should investigate whether the opposite changes in the numbers of D2 and A2A receptors in dystonic mice could also compromise the formation of D2/A2A etheromers, changing their interaction not only quantitatively but also qualitatively.

### 3.3. Changes in Second Messenger cAMP

To verify whether the tissue levels of cAMP are influenced by the changes in A2A expression, we evaluated the basal content of cAMP in tissue homogenates of the different basal ganglia nuclei from Tor1a+/+ and Tor1a+/− knock-out mice. Surprisingly, we found significantly increased cAMP levels only in the striatum of Tor1a+/−, whereas in the globus pallidus and in the entopeduncular nucleus, cAMP levels were similar in Tor1a+/+ and in Tor1a+/− mice. The cAMP synthesis in striatal neurons is stimulated by dopamine D1 receptors, which are significantly decreased in DYT1 transgenic mice [36], and by A2A receptors, which are significantly increased in Tor1a+/− mice (present work), and so likely determining the increased striatal cAMP levels in the Tor1a+/− striatum.

To explain the discrepancy between the higher levels of cAMP in the Tor1a+/− striatum and the unchanged cAMP levels in the globus pallidus and entopeduncular nucleus, we should consider the interference of other related factors in addition to the increased or decreased stimulating action of the A2A receptors. We can hypothesize that the A2A receptor dependent cAMP synthesis is out of the inhibitory control of the deficient D2 receptors, but the D2 receptors appear to be functionally impaired both in the striatum as well as in the globus pallidus [8,32].

In a previous work, we reported that PDE10A-dependent cAMP hydrolyzing activity as well as total PDE activity are unaffected in the striatum of a mutant mouse model of DYT1 dystonia, wherein at the same time the PDE10A-dependent cAMP hydrolyzing activity was increased in the globus pallidus and decreased in the entopeduncular nucleus [37]. Therefore, we could hypothesize that in the mutant DYT1 striatum, an insufficient total PDE activity cannot timely catabolize the A2A dependent increased cAMP synthesis, whereas the equilibrium between cAMP synthesis and catabolism is preserved in the globus pallidus and in the entopeduncular nucleus by a contemporary increase or decrease in PDE activity, respectively. The reasons for the discrepant compensatory PDE activity in front of A2A-dependent cAMP synthesis in the striatum of mutant DYT1 mice are unknown. We can just speculate on the different cellular distributions of the PDE10A isoform in different striatal neurons.

Indeed, PDE10A is selectively expressed in medium spiny neurons, but not in striatal interneurons, and in particular it is absent in cholinergic interneurons [38,39]. A2A receptors on cholinergic interneurons play a crucial role in striatal network function, activating the release of acetylcholine in the striatum, and striatal cholinergic dysfunction has been shown to have a widespread role in the pathophysiology of dystonia [21,40]. Therefore, we can hypothesize that the increased levels of cAMP in the striatum may reflect, at least in part, the A2A-dependent cAMP synthesis in cholinergic interneurons, wherein cAMP is not hydrolysable by the absent PDE10A, nor adequately hydrolyzed by another unknown phosphodiesterase isoform.

Anyway, the relevant impact of cAMP on the membrane excitability of striatal medium-sized spiny neurons and cholinergic interneurons has been widely described [41,42,43]. Our observation of a chronic elevation in neuronal cAMP in the striatum of Tor1a+/− can open a new window to investigate the role of cAMP changes in the severely altered synaptic plasticity of medium spiny neurons and cholinergic interneurons in dystonia mouse models [8,11,12,13,14,15,16,40,44].

### 3.4. Changes in A2A mRNA

Finally, to evaluate whether the changes in A2A expression in Tor1a+/− mice are regulated at translational or post-translational levels, we used the sensitive real-time PCR technique analysis of A2A mRNA in basal ganglia. According to previous studies, the A2A mRNA can be found only on the striatum in cell bodies of medium spiny striatal neurons and occasionally in dendrites, coding for the synthesis of A2A receptors, which would be transported to the terminals of striatal-pallidal efferent neurons till the globus pallidus [45]. Within the basal ganglia, we detected significant levels of A2A mRNA only in the striatum of Tor1a+/+ that were similar to A2A mRNA levels in the striatum of Tor1a+/−, suggesting that the increased A2A expression in the Tor1a+/− striatum can be regulated at post-translational levels. However, the striatal results of A2A mRNA should be kept with caution since in tissue homogenates we cannot distinguish eventual differentiated A2A mRNA levels in different classes of striatal neurons either to the direct or indirect pathway, obtaining on the whole apparently unchanged striatal A2A mRNA levels. 

Instead, a surprising A2A mRNA significant increase was observed in the globus pallidus of Tor1a+/−, whereas A2A mRNA was detected just in a very low amount in the globus pallidus of Tor1a+/+. Therefore, in Tor1a+/− it is likely that the globus pallidus is coding for its own A2A receptors, and such A2A neo-expression may change, at least in part, its dependent role from the striatal A2A inputs. The A2A mRNA changes in the basal ganglia of Tor1a+/− appear to be specific, since in Parkinson’s disease A2A mRNA appears either decreased in the striatum or unchanged in the globus pallidus [46]. Future studies should clarify the cell types expressing A2A mRNA in the globus pallidus of Tor1a+/− mice, and the function of their intrinsic A2A receptors.

## 4. Materials and Methods

### 4.1. Animals 

C57BL/6 Tor1a+/− knock-out mice, which mimic the loss of function of the DYT1 dystonia *Tor1a* mutation [47], were bred at Santa Lucia Foundation Animal Facility. Control Tor1a+/+, and Tor1a+/− knock-out mice were kept under an artificial day-night cycle, with free access to food and water. Male control and mutant mice were sacrificed at the age of 5–6 months. DNA was isolated and amplified from 1 to 2 mm tail fragments with the Extract-N-Amp Tissue polymerase chain reaction (PCR) kit (XNAT2 kit; Sigma-Aldrich Merck Life Science, Milan, Italy), and genotyping was performed as previously reported [17]. All the efforts were made to minimize the number of animals utilized and their suffering. Treatment and handling of mice were carried out in compliance with both the European Council and Italian guidelines (2010/63EU, 20 October 2010; D.L. 26/2014, 4 March 2014; 86/609/EEC, 24 November 1986; D.L. 116/1992—27 January 1992), according to experimental protocols approved by the Animal Ethics Committee of the University of Rome Tor Vergata (D.M. 153/2001-A, 9 April 2001, and 43/2002-A, 4 May 2002) and by the IRCCS Santa Lucia Foundation Animal Care and Use Committee (D.M.9/2006-A, 2006), and authorized by the Italian Ministry of Health (authorization 223/2017-PR, May 2017).

For biochemical studies, the animals were killed by cervical dislocation, and the brains were removed rapidly and placed on an ice-cold plate. Thick brain sections were cut with an Oxford vibratome, and the caudate–putamen, globus pallidus and entopeduncular nucleus were dissected out rapidly from both hemispheres under a stereomicroscope, and promptly frozen in liquid nitrogen and stored at −80 °C [48].

For morphological studies, the animals were deeply anesthetized with tiletamine/zolazepam (80 mg/kg) and xylazine (10 mg/kg) and perfused trans-cardially with 1% heparin in 50 mL 0.1 M sodium phosphate buffer (PBS), and with 250 mL 4% para-formaldehyde in 0.1 M in PBS (pH 7.4). The brains were removed immediately and post-fixed in the same fixative solution overnight at 4 °C; then they were equilibrated with 30% sucrose overnight, and finally frozen in liquid nitrogen and stored at −80 °C [48].

### 4.2. Quantitative Analysis of A2A Protein

The quantitative analysis of A2A expressions in basal ganglia was assessed by Western blotting. Tissues were lysated in 20 mM Tris-HCl buffer pH 7.2 containing 0.2 mM EGTA, 5 mM MgCl2, 0.1% *v*/*v* Triton X-100, 5 mM β–mercaptoethanol, 1 mM PMSF and 2% *v*/*v* antiprotease cocktail (Sigma–Aldrich Merck Life Science, Milan, Italy). Thirty micrograms of proteins were loaded on a 9% SDS polyacrylamide gel and subjected to electrophoresis under a reducing condition. The proteins were then transferred to a nitrocellulose membrane (Bio–Rad, Milan, Italy). The blots were incubated overnight at 4 °C with a rabbit polyclonal anti–A2A receptor antibody (1:1000; BML–SA654, Enzo Life Sciences, Farmingdale, NY, USA), or mouse anti-β-actin (1:10,000; Sigma-Aldrich Merck Life Science, Milan, Italy) as a reference standard. A2A receptors-reactive bands were revealed by horseradish peroxidase-conjugated secondary antibodies (1:10,000, Jackson Immunoresearch, Cambridgeshire, UK), incubated in a lumi-light-enhanced chemiluminescence substrate (Bio-Rad, Milan, Italy), and exposed to Chemidoc (Bio–Rad, Milan, Italy). Densitometric analysis of scanned blots was performed using the NIH ImageJ version l.29 program (NIH, Bethesda, MD, USA).

### 4.3. cAMP Measurement

The frozen samples were immediately submerged in 0.1 M HCl, and rapidly homogenized. The homogenates were centrifuged at 13,000× *g* for 30 min at room temperature and the supernatants were acetylated, according to the instructions of the commercial kit manufacturer (Direct EIA kit, Enzo Life Sciences, Farmingdale, NY, USA). The kit uses a polyclonal antibody to cAMP, which in a competitive manner binds the cyclic nucleotides in the sample, or cyclic nucleotides conjugated with alkaline phosphatase molecules added to the incubation medium. The samples were incubated for 2 h at room temperature, and thereafter the substrate p-nitrophenyl phosphate (pNPP) was added for 1 h to reveal the residual alkaline phosphatase activity in the medium, which generates a yellow product, readable on a microplate reader (Multiskan™ FC Microplate Photometer, Thermo Fisher Scientific, Monza, Italy) at 405 nm. The intensity of the yellow color in the medium is inversely proportional to the concentration of cyclic nucleotides in the samples and in comparative standards.

### 4.4. RNA Extraction and Real–Time PCR

Total RNA was prepared using the TRI Reagent (Sigma-Aldrich Merck Life Science, Milan, Italy), according to the manufacturer’s instructions, and quantified with a NanoDrop 1000 spectrophotometer (Thermo Fisher Scientific, Monza, Italy). One microgram of RNA was reverse-transcribed using the QuantiTect reverse transcription kit (Qiagen, Milan, Italy). Real-time PCR was performed on reverse-transcription products with the PowerUp™ SYBR™ Green Master Mix in the QuantStudio 3 Real-Time PCR System (Thermo Fisher Scientific, Monza, Italy). Thermal cycling conditions consisted of an initial denaturation step at 95 °C for 2 min, followed by 40 cycles at 95 °C for 15 s, 60 °C for 15 s, and 72 °C for 1 min. The threshold cycle (Ct) (defined as the fractional PCR cycle number at which fluorescence reaches 10 times the baseline SD) was used for comparison analysis. The 2^−ΔΔCt^ method was used to evaluate the relative expression ratio for A2A receptors compared with the internal control gene β–actin. The following sequences of primers were used: A2A receptors (a) forward 5′–TCTTCTTCGCCTGCTTTGTCC–3′, (b) reverse 5′–GCCCTCATACCCGTCACCA–3′; β-actin (a) forward 5′–GCGCAAGTACTCTGTGTGGA–3′, (b) reverse 5′–AAGGGTGTAAAACGCAGCT–3′.

### 4.5. A2A Immunofluorescence in Confocal Microscopy 

Coronal brain sections (40 μm thick) were cut with a freezing microtome. Tissue sections were then incubated for 1 h at room temperature in 10% donkey serum solution in PBS 0.25%–Triton X-100 (PBS-Tx). The sections were incubated with the primary rabbit anti-A2A antibody (BML–SA654, Enzo Life Sciences, Farmingdale, NY, USA) 1:200, 3 days at 4 °C; thereafter, the sections were incubated with cyanine 3 (cy3) conjugated secondary antibody (Jackson Immuno Research, Cambridgeshire, UK) 1:200, 2 h at room temperature. For negative controls, representative sections were processed with the omission of the primary or of the secondary antibody. Cell nuclei were detected with a blue-fluorescent DNA stain by 4′,6–diamidino–2–phenylindole (DAPI) (D9542, Sigma-Aldrich Merck Life Science, Milan, Italy). After washout, tissue sections were mounted on plus polarized glass slides with Vectashield mounting medium (Super Frost Plus, Thermo Fisher Scientific, Monza, Italy) and cover-slipped.

Fluorescent images were acquired with a LSM700 Zeiss confocal laser scanning microscope (Zeiss, Oberkochen, Germany), with a 5×, a 20× objective, or a 63× oil immersion lens (1.4 numerical aperture) with an additional digital zoom factor (1×–1.5×–2×) under no saturation conditions. Single-section images (1024 × 1024) or *z*-stack projections in the *z*-dimension (z-spacing, 1 μm) were collected. Z-stack images were acquired to analyze the whole neuronal soma, which spans multiple confocal planes. The confocal pinhole was kept at 1, the gain and the offset were adjusted to prevent saturation of the brightest signal and sequential scanning for each channel was performed. The confocal settings, including laser power, photomultiplier gain, and offset, were kept constant for each marker. 

A negative staining was obtained omitting the primary or the secondary antibody. The 5× and 20× objectives were used to define areas of interest in the dorsolateral striatum, globus pallidus, and entopeduncular nucleus; the distribution of A2A receptors was first acquired using 20× objective with an additional digital zoom factor (1×–1.5×–2×), and thereafter 63× oil immersion objective (1.4 numerical aperture).

For quantitative analysis of the number of A2A positive spots per microscopic field and of their perimeters, images were collected from at least 3–4 slices per animal, processed simultaneously from each basal ganglia nucleus (*n* ≥ 4 mice/genotype), and exported for analysis with ImageJ software (NIH, Bethesda, MD, USA). Software background subtraction was utilized to reduce noise.

### 4.6. A2A Localization in Striatal Cholinergic Interneurons 

A double immunofluorescent staining for choline-acetyltransferase (ChAT) and A2A was used to evaluate the colocalization of A2A receptors in striatal cholinergic interneurons. Brain sections were incubated with goat anti-ChAT (Nova biological, CA, USA) and rabbit anti-A2A receptors (BML–SA654, Enzo Life Sciences, Farmingdale, NY, USA). Both primary antibodies were used at a 1:200 dilution, in 0.1 M PBSS containing 0.3% Triton X–100 for 72 h at 4 °C. Sections were then rinsed three times for 5 min at room temperature and subsequently incubated with cyanine 2 (Cy2) and cyanine 3 (cy3) conjugated secondary antibodies (Jackson Immuno Research, Cambridgeshire, UK) for 2 h at room temperature at 1:200 dilution in a 0.1 M PB solution containing 0.3% Triton X–100. For negative controls, representative sections were processed with the omission of the primary or of the secondary antibodies. Cell nuclei were detected with a blue-fluorescent DNA stain DAPI (D9542, Sigma-Aldrich Merck Life Science, Milan, Italy). Subsequently, sections were rinsed in PBS, mounted on plus polarized glass slides with Vectashield mounting medium (Super Frost Plus, Thermo Fisher Scientific, Monza, Italy) and cover-slipped.

The sections were preliminary examined under an epi-illumination fluorescence microscope (Zeiss Axioskop 2, Oberkochen, Germany). Confocal laser scanner microscopy (Zeiss LSM800, Oberkochen, Germany) was used to acquire immunofluorescent images as reported in the previous subsection. 

Immunofluorescence intensity and colocalization analysis were evaluated by using the Java image processing and plugin analysis program included in Fiji ImageJ (NIH, Bethesda, MD, USA). To quantify the density of the A2A specific marker on a defined area, the “overlapping signal” of the A2A signal on ChAT neuronal bodies was calculated.

### 4.7. Experimental Design and Statistical Analysis

Biological samples and microscopic fields were randomly selected from the basal ganglia areas of the two experimental groups. All data were initially entered into an Excel database (Microsoft, Redmond, WA, USA) and the analysis was performed using the Statistical Package for the Social Sciences Windows, version 15.0 (SPSS, Chicago, IL, USA). Descriptive statistics consisted of the mean ± standard deviation (SD) for parameters with gaussian distributions (after confirmation with histograms and the Kolgomorov–Smirnov test), or of the mean ± standard error mean (SEM). Comparisons were performed in compliance with data characteristics either with the Paired Samples Test, or ANOVA one-way or ANOVA two factors and multiple comparisons by Bonferroni test. A *p* value of < 0.05 was considered statistically significant.

## 5. Conclusions

The increased expression of A2A receptors in the striatal-pallidal complex of Tor1a+/− knock-out mice may suggest a gaining of function of such receptors, overwhelming the D2 loss of function in the indirect pathway; moreover, the associated decrease in A2A expression in the entopeduncular nucleus may be functionally significative. Future studies should clarify the relevance of A2A changes in the direct and indirect pathway in dysregulating the basal ganglia network in dystonia pathophysiology.

## Figures and Tables

**Figure 1 ijms-22-02691-f001:**
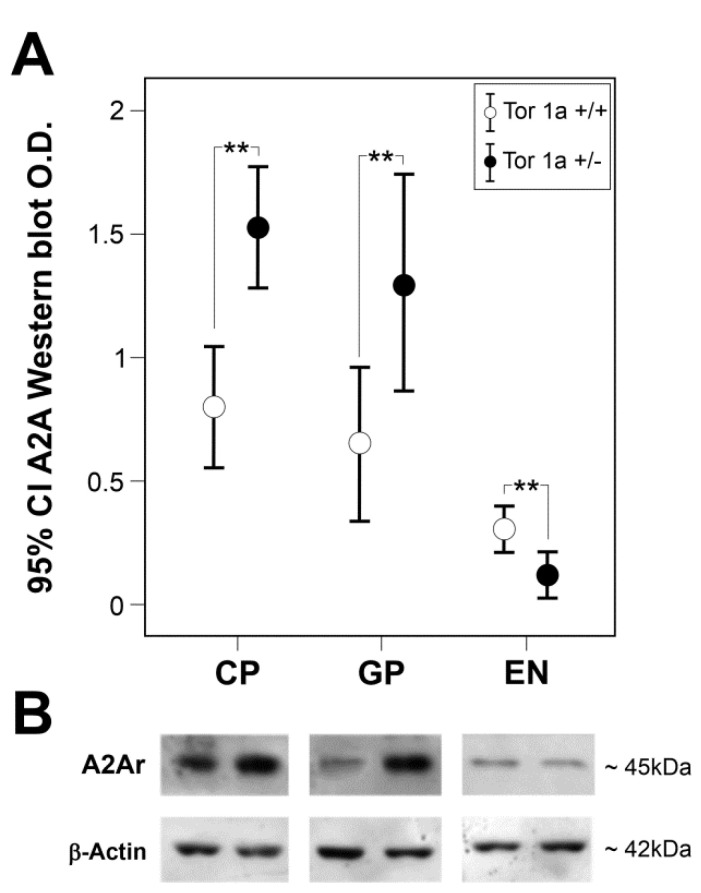
Expression of A2A in basal ganglia of Tor1a+/+ and Tor1a+/− mice. (**A**) Densitometric analysis optical density (O.D.) of A2A-immunostained bands in correspondent areas of caudate–putamen (CP), globus pallidus (GP), and entopeduncular nucleus (EN) of Tor1a+/+ and Tor1a+/−. Results were expressed as the mean ± SD of the values obtained in each group. One-way ANOVA, ** *p* < 0.020. (**B**) Representative immunoblots of A2A content in correspondent area of CP, GP and EN of Tor1a+/+ and Tor1a+/− mice. As an internal reference standard, the β-actin content was detected in each lane of the same blots to correct for protein loading.

**Figure 2 ijms-22-02691-f002:**
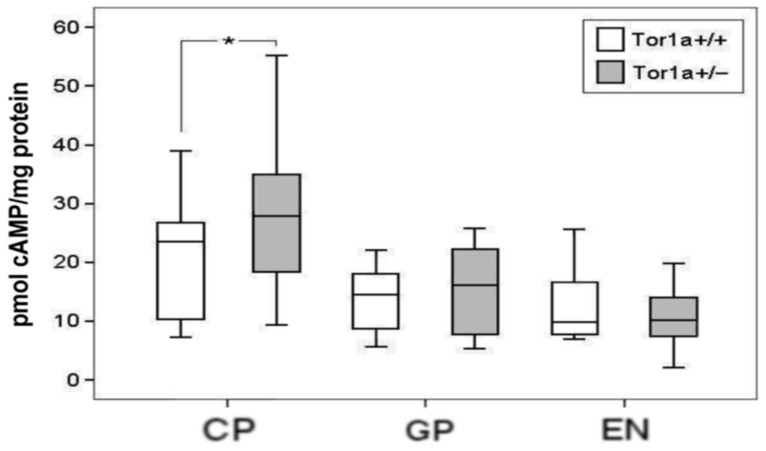
Cyclic adenosine monophosphate (cAMP) level in tissue homogenates of caudate–putamen (CP), globus pallidus (GP), and entopeduncular nucleus (EN) in control Tor1a+/+ and in mutant Tor1a+/− mice. Values, expressed as pmol of cAMP/ mg protein, are represented as mean ± SD. Significant increase in cAMP content was detected in the CP of Tor1a+/− versus Tor1a+/+. One-way ANOVA, * *p* < 0.05.

**Figure 3 ijms-22-02691-f003:**
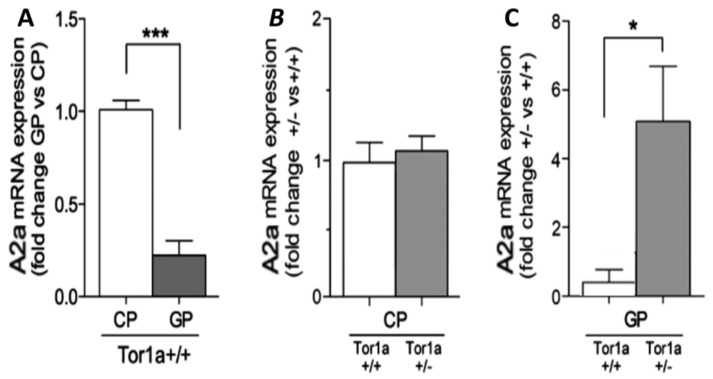
(**A**)—Comparative real-time PCR of A2A mRNA in caudate–putamen (CP) and globus pallidus (GP) of Tor1a+/+ control mice (*n* = 6, df = 1, F = 71.172, *p* = 0.001). (**B**)—The expression of A2A mRNA in the CP of Tor1a+/+ mice is compared to the levels in the CP of Tor1a+/− mice (*n* = 8, df = 1, F = 0.311, *p* = 0.586). (**C**)—The expression of A2A mRNA in the GP of Tor1a+/+ mice is compared to the levels in the GP of Tor1a+/− mice (*n* = 7, df = 1, F = 4.743, *p* = 0.048). Results were expressed as the mean ± SD of values obtained from each group. Data in (**A**–**C**) were analyzed with one-way ANOVA followed by Bonferroni’s correction for multiple data. * *p* < 0.05, *** *p* < 0.001

**Figure 4 ijms-22-02691-f004:**
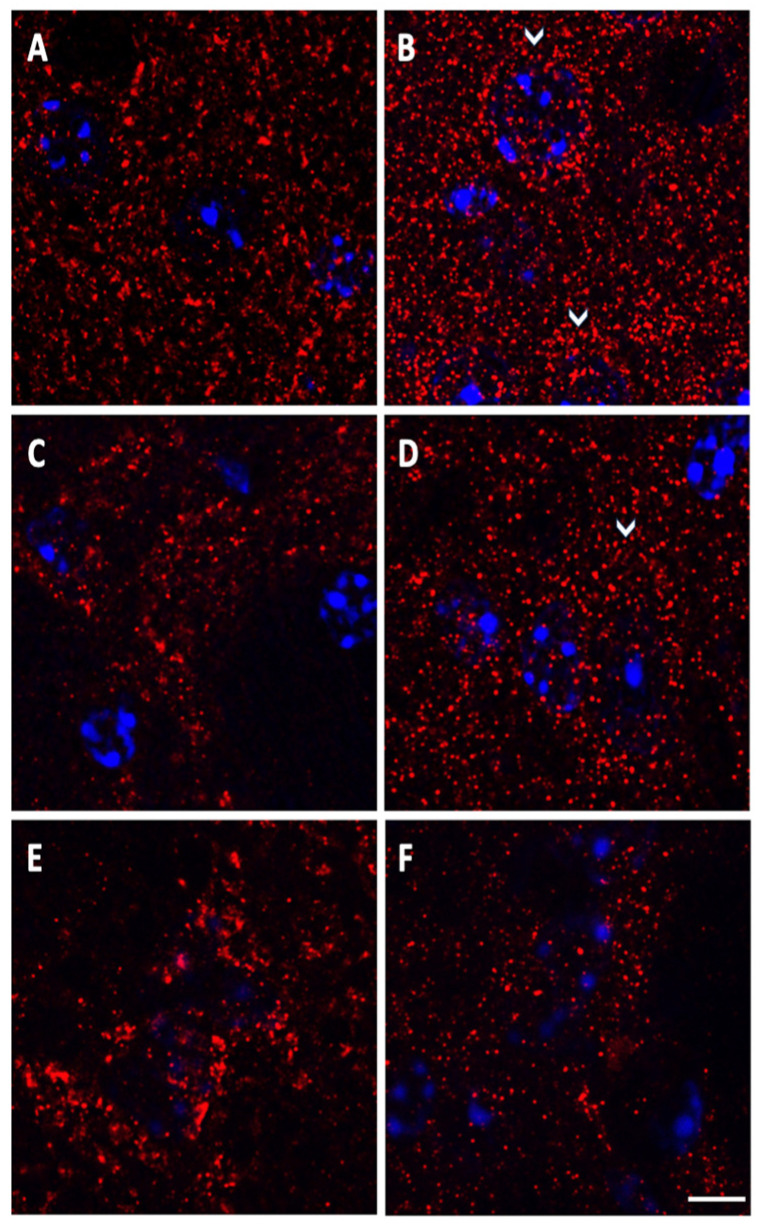
Representative immuno-fluorescence microphotographs of high magnification (63× oil immersion objective plus zoom factor 1×–1.5×–2.5×) confocal laser scanning microscopy, showing the roundish morphology of A2A positive tiny spots and their distributions in the striatum (**A**,**B**), globus pallidus (**C**,**D**), and entopeduncular nucleus (**E**,**F**) of control Tor1a+/+ (**A**,**C**,**E**), and of mutant Tor1a+/− (**B**,**D**,**F**) mice. A2A receptor labeling is visualized in red-Cy3 fluorescence, while cell nuclei are visualized by 4′,6–diamidino–2–phenylindole (DAPI) fluorescence in blue. White arrows in B and D point to clustering of A2A grains around neuronal bodies. Scale Bar in F = 5 µm.

**Figure 5 ijms-22-02691-f005:**
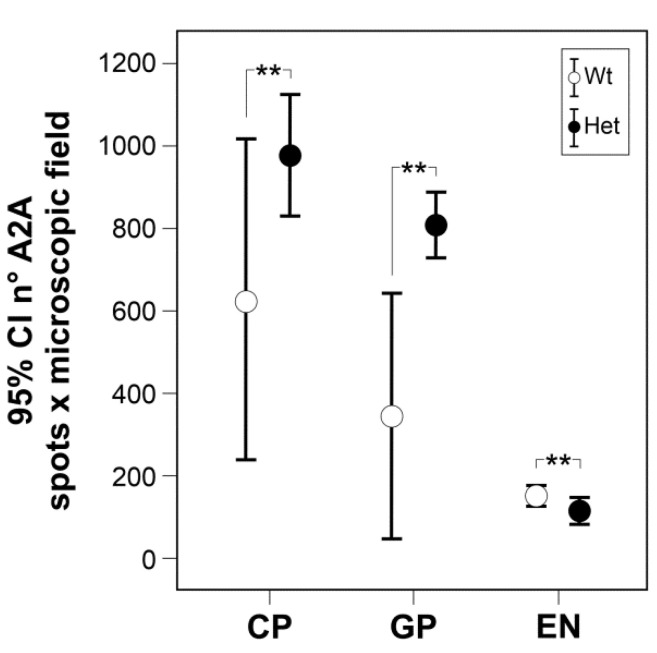
Comparative densitometric analysis of the number (n°) of A2A positive grains per microscopic field in caudate–putamen (CP), globus pallidus (GP), and entopeduncular nucleus (EN) of control Tor1a+/+ and of mutant Tor1a+/− mice. One-way ANOVA, ** *p* < 0.02.

**Figure 6 ijms-22-02691-f006:**
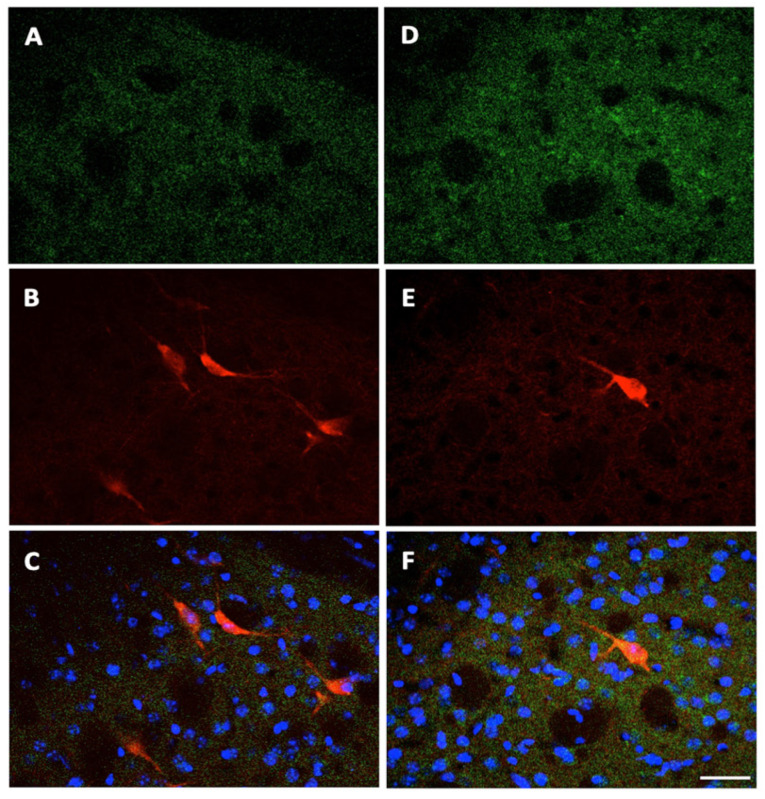
Localization of A2A receptors in cholinergic striatal interneurons. Representative confocal laser scanning microscopy images double-labeled with antibody for anti A2A receptors and with antibody anti-ChAT for cholinergic interneurons. A2A receptors are visualized in green-cy2 fluorescence (**A**,**D**); ChAT positive neurons are visualized in red-cy3 fluorescence (**B**,**E**); merged images in C, F with A2A positive grains visualized in yellow on ChAT-positive interneurons, and in green in the neuropil. Cell nuclei are visualized by DAPI fluorescence in blue. (**A**–**C**) Tor1a+/+ mice; (**D**–**F**) Tor1a+/− mice. Scale bar in F = 50 μm.

## Data Availability

Data and methods used in the research have been presented in sufficient detail in the paper. Additional data can be requested to the corresponding author.

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
