# Peer review of "A2A Receptor Dysregulation in Dystonia DYT1 Knock-Out Mice"

_ijms, 2021, doi:10.3390/ijms22052691_

Round 1
Reviewer 1 Report
Thanks to the authors for providing better figure 4 and an additional figure 6 together with a quantification of figure X in the cover letter. They also give a sincere response with the reasons why they cannot perform the additional suggested experiments, which I truly understand.
They clarified all the major and minor points putting new and clearer information in the results part, homogenizing the statistical test and suggesting new interesting ideas in the results part.
Reviewer 2 Report
After revision, the paper is acceptable for publication, although there are still typing errors in the text. Figure 6 legend does not indicate origin of samples and image quality could be improved.
This manuscript is a resubmission of an earlier submission. The following is a list of the peer review reports and author responses from that submission.
Round 1
Reviewer 1 Report
The study by D’Angelo et al describes the distribution of the A2A receptor protein and mRNA, in basal ganglia of a mouse model of dystonia.
Although the study is interesting and may potentially offer new insights in the field, it requires additional experimental analysis. As the authors discuss, they could in particular test the localization of the receptor on synaptosomes, or on pre- and post-synaptic fractions.
It would also be useful to present control markers for the different brain areas in Western blots.
Specific points:
Please also show dopamine D2 receptor’s localization in this model, and compare to A2A receptor’s distribution (by IF).
Fig. 4B: please provide a better image (the red channel level is not comparable to the one in Fig. 4A)
Fig. 6: The images present too saturated color levels; please provide comparative images of separated and merged color channels.
Note: probably due to tech problems during the PDF conversion several different characters were introduced in various sections (e.g., in M&M, rows 407-411)
Reviewer 2 Report
In the present manuscript D´Angelo et al., described in a series of experiments changes in the expression of the A2A receptor in different brain areas of a Tor1a +/- mice. The experiments are well design and they use proper controls for their experiments. They demonstrate an increase of this receptor in the CP and GP of Tor1a +/- mice and a decrease in the EN. They also observed an increase cAMP in the CP.
Overall the study is well designed and executed, there are, however some major and minor concerns listed below.
Major points:
- The authors claim that A2A is in the synapse although they cannot detect it clear. For showing the localization of the receptor in the synapse a double immunostaining with a synaptic marker should be performed and colocalization analyzed. Another option could be to perform a subcellular fractionation of the synapsis and analyze A2A by WB in the synaptic fraction.
- The authors hypothesize that GABAergic inhibitory inputs at SP level could be increase. Some GABAergic markers should be check to confirm it for example GAD65 or GAD67, or calcium-binding proteins or some neuropeptids co-expressed in GABAergic neurons.
- In results part 2.5 and figure 6 it is not clear what is what. Is it figure A Tor1a +/+ and figure B Tor1a +/- ? It should be better explained and also clarified in the figure legend.
- Line 128: how do you recognize the striatal axonal bundles in figure 4?
Minor points:
- Line 72: include after EN of Tor1a +/+ and Tor1a +/-
- Figure 1: Are the samples from the same gel? Can you compare the different brain areas between them?
- Different statistic tests are used in fig 1, 2 and 3 but you are always just comparing the 2 mice. Statistical analysis should be uniformed.
- By convention α of the p-value is set as 0.05, 0.01, 0.001 and 0.0001. In figure 1, the authors used **p ≤ 0.02. This part should be more clarified in the materials and methods.
Round 2
Reviewer 1 Report
In order to render the ms acceptable for publication, the authors should at least perform subcellular fractionation analysis to verify the receptor's localization in the synapse.
Furthermore, the quality of images has not been improved (Fig. 4B and Fig. 6).
In its present form, I consider the ms not acceptable for publication.
Reviewer 2 Report
The authors answer to all the questions and justified or clarify in the text the need of performing more experiments (as the suggested stainings) in order to further corroborate their results.